# Predicting treatment outcomes following an exacerbation of airways disease

Andreas Halner [1], Sally Beer[2], Richard Pullinger[2], Mona Bafadhel[1], Richard E. K. Russell [1,3]*

1 Respiratory Medicine Unit, Nuffield Department of Medicine, University of Oxford, Oxford, United Kingdom, 2 Department of Emergency Medicine, Oxford University Hospitals NHS Foundation Trust, Oxford, United Kingdom, 3 NIHR Biomedical Research Centre, University of Oxford, Oxford, United Kingdom

* Richard.Russell@ndm.ox.ac.uk

## Abstract

### Background

COPD and asthma exacerbations result in many emergency department admissions. Not all treatments are successful, often leading to hospital readmissions.

### Aims

We sought to develop predictive models for exacerbation treatment outcome in a cohort of exacerbating asthma and COPD patients presenting to the emergency department.

### Methods

Treatment failure was defined as the need for additional systemic corticosteroids (SCS) and/or antibiotics, hospital readmission or death within 30 days of initial emergency department visit. We performed univariate analysis comparing characteristics of patients either given or not given SCS at exacerbation and of patients who succeeded versus failed treatment. Patient demographics, medications and exacerbation symptoms, physiology and biology were available. We developed multivariate random forest models to identify predictors of SCS prescription and for predicting treatment failure.

### Results

Data were available for 81 patients, 43 (53%) of whom failed treatment. 64 (79%) of patients were given SCS. A random forest model using presence of wheeze at exacerbation and blood eosinophil percentage predicted SCS prescription with area under receiver operating characteristic curve (AUC) 0.69. An 11 variable random forest model (which included medication, previous exacerbations, symptoms and quality of life scores) could predict treatment failure with AUC 0.81. A random forest model using just the two best predictors of treatment failure, namely, visual analogue scale for breathlessness and sputum purulence, predicted treatment failure with AUC 0.68.

**Data Availability Statement:** All relevant data are within the manuscript and its Supporting Information files.

**Funding:** The research was supported by the National Institute for Health Research (NIHR)

Oxford Biomedical Research Centre (BRC). The views expressed are not necessarily those of the NHS, the NIHR or the Department of Health. The NIHR/NHS/Department of Health had no role in study design, data collection and analysis, decision to publish or preparation of the manuscript.

**Competing interests:** Mona Bafadhel reports outside the submitted work research grant reports from AZ; honoraria from AZ, Chiesi, and GlaxoSmithKline; and is on the scientific advisory board for AlbusHealth® and ProAxsis®. Richard Russell has received honoraria from AZ, GSK, Boheringer Ingelheim, Chiesi, Cipla and is on the advisory board for AlbusHealth®, has received research funding from Circassia UK and his work is supported by the Oxford NIHR Biomedical Research Centre. The remaining authors have declared that no competing interests exist. This does not alter our adherence to PLOS ONE policies on sharing data and materials.

## Conclusion

Prediction of exacerbation treatment outcome can be achieved via supervised machine learning combining different predictors at exacerbation. Validation of our predictive models in separate, larger patient cohorts is required.

## Introduction

Exacerbations of COPD and asthma are disruptive to patients' lives, are associated with increased risk of future exacerbations, increased mortality risk and are a large economic burden; furthermore, each event may require hospital admission [1–3]. Not all patients improve following standard initial treatment (as per NICE guidelines with SCS and/or antibiotics [2, 4]) and thus may require additional treatment with systemic corticosteroids (SCS) and/or antibiotics or even hospital readmission [1, 2]. Although part of guidance, studies demonstrate inconsistent benefits for SCS and/or antibiotics for treating exacerbations of COPD and asthma in which treatment is received in hospital [5–8], with increasing concern that SCS causes harm [9]. Single centre [10] and multi-centre studies [11] have examined the use of the peripheral blood eosinophil at the time of an exacerbation to direct SCS use in patients with COPD exacerbation, but little is known about why some patients admitted to hospital succeed whereas others fail treatment. There is thus an urgent need to develop predictive tools to identify which patients at exacerbation will respond successfully to treatment and which patients will not. Such a tool could help identify patients at high risk of treatment failure who require closer monitoring following their presentation to the emergency department (ED), as well as developing biomarker-driven treatment algorithms to optimise patient response to treatment. Since ED presentations are not treated by pulmonologists, a tool to help guide treatment of response in a personalised medicine way is needed for specialists and non-specialists.

In this study, we investigate factors, in patients that attend ED with an exacerbation of asthma or COPD, which may be associated with treatment failure and physician treatment prescription decision. We then use a data-driven approach to develop a multivariate supervised learning algorithm to predict treatment failure of asthma and COPD patients admitted with an exacerbation, as well as to predict the physician decision to prescribe SCS at exacerbation.

## Methods

### Participants

Participants were recruited if they had a patient-reported primary or secondary care diagnosis of asthma or COPD and presented to the ED of the John Radcliffe Hospital, a large teaching hospital in Oxfordshire, with an exacerbation of COPD or asthma. Exacerbations of COPD and asthma were defined and treated as per GOLD [12] and BTS respectively [13]. Participants with a current history of active pulmonary tuberculosis or current primary malignancy or with any other clinically relevant lung disease judged to be the primary diagnosis were excluded. Any alternative causes for non-exacerbation related increase in symptoms were also excluded, including but not limited to pneumonia, pulmonary embolism, pneumothorax or primary ischemic event. All participants provided informed written consent and the study was approved by the North West London Research Ethics Committee (REC: 15/LO/2119).

## Study design

Observational and routinely collected clinical data were prospectively collected for a study duration of 12 months. Study measurements were made on the day of exacerbation termed D0 and on the 1st, 5th day (or day of discharge) and 30th day after exacerbation termed D1, D5 and D30.

## Measurements

Data collection included demographics and medication history as well as medical history data such as past asthma or COPD diagnosis. Participant symptoms, health status and quality of life were assessed using participant reported outcome measures including Visual Analogue Scale (VAS) [14], Medical Research Council dyspnoea scale [15], Hospital Anxiety and Depression scale [16] and EuroQol 5D [17]. Physiological measurements taken were the resting oxygen saturations, heart rate and respiratory rate. Venous blood samples were collected and analysed to characterise participants' inflammatory phenotype; these measurements included peripheral blood eosinophils, C-reactive protein (CRP), biochemistry and glucose.

## Statistical analyses

Only participants with complete D30 follow-up data were considered for statistical analysis performed using the programming language 'R' [18]. Numerical data were first assessed for normality using the shapiro-wilk test. Normally-distributed data are shown as mean (range) whereas non-normally distributed data are shown as median (interquartile range (IQR)). Treatment failure was defined as a new hospital readmission and/or new and additional treatment (antibiotics and/or SCS) or death between the day of discharge and D30. For numerical variables, the t-test or wilcox test were used for comparing two groups depending on whether the data were normally or non-normally distributed respectively. Data transformations to non-normally distributed variables was not performed. When comparing more than two groups, either the one-way anova test (and Tukey post hoc test) or the kruskal-wallis test (and post-hoc Kruskal Dunn test) was used, depending on whether the normality assumption was met. The chi-squared test was used for binary or categorical data. The relationship between exacerbation VAS symptoms and peripheral blood biomarkers was calculated using Spearman's rho ($r_s$). Exploratory univariate analysis of variables at the time of exacerbation was performed to identify differences in characteristics of participants based on exacerbation treatment and identification of variables at D0 variables which differ between treatment failure versus treatment success. Multivariate random forest models were drawn up to predict treatment failure (defined as above) and SCS clinical prescription (see S1 Appendix for more details).

The random forests were then trained on subsets of the variables as part of a feature elimination procedure with leave-one-out cross validation which allows validation of the model performance (see S1 Appendix for full statistical methods) [19, 20].

## Results

Out of 104 participants who entered the study, there were 83 with D30 follow-up data. Complete data were available for analysis in 81 participants. A diagnosis of asthma and COPD was found in 59 (73%) and 22 (27%) participants respectively. Treatment allocation showed that at the time of an exacerbation 31 participants (38%) received both SCS and antibiotics; 33 participants (41%) received SCS alone; 6 participants (7%) received antibiotics alone; and 11 participants (14%) received neither. The demographics and at exacerbation characteristics of the asthma and COPD participants are shown in **Table 1**. The participants with COPD were older

**Table 1. Characteristics of the asthma and COPD participants at exacerbation.**

| Characteristic | Primary Respiratory Diagnosis | | P-value |
|---|---|---|---|
| | Asthma (n = 59) | COPD (n = 22) | |
| Male, n (%) | 20 (34) | 14 (64) | 0.02 |
| Current smokers, n (%) | 15 (25) | 6 (27) | <0.001 |
| Ex-smokers, n (%) | 18 (31) | 16 (73) | |
| Never smoker, n (%) | 26 (44) | 0 (0) | |
| Number taking ICS, n (%) | 33 (56) | 7 (32) | 0.05 |
| Increased wheeze at exacerbation, n (%) | 51 (86) | 19 (86) | 0.99 |
| Age (years) | 41 (28–55) | 67 (60–72) | <0.001 |
| Pack year history | 0.3 (0.0–12.4) | 33.9 (16.0–52.5) | <0.001 |
| Number of hospital admissions in previous 12 months | 0 (0–1) | 0 (0–1) | 0.68 |
| Oxygen saturation (%) | 95 (94–97) | 93 (91–95) | <0.001 |
| VAS cough (mm) | 55 (26–72) | 42 (25–53) | 0.22 |
| VAS breathlessness (mm) | 70 (29–84) | 76 (44–97) | 0.20 |
| VAS sputum production (mm) | 19 (2–48) | 21 (1–41) | 0.96 |
| VAS sputum purulence (mm) | 5 (0–48) | 23 (0–53) | 0.29 |
| Leucocytes, ($\times 10^9$cells/L) | 10.8 (8.9–12.2) | 10.7 (9.1–15.5) | 0.25 |
| Neutrophils ($\times 10^9$cells/L) | 7.9 (5.4–9.5) | 7.9 (5.9–12.2) | 0.19 |
| Eosinophils ($\times 10^9$cells/L) | 0.2 (0.0–0.4) | 0.1 (0.0–0.3) | 0.65 |
| CRP (mg/L) | 7.2 (1.7–23.6) | 17.3 (1.8–58.1) | 0.14 |

*Definition of abbreviations*: ICS = inhaled corticosteroids; CRP = C-reactive protein.
*Measures of central tendency and spread*: all variables are presented as median (IQR) or as number (%) of instances. Comparisons between asthma and COPD are made using chi-squared or wilcox test as appropriate. P-values are reported to 2dp unless the p-value is less than 0.001, in which case it is reported as <0.001.

(median (IQR) age 67 (60–72) years compared to participants with asthma 41 (28–55) years, p<0.01) and at exacerbation had a lower % oxygen saturation (median oxygen saturation in COPD 93% versus median in asthma 95%, p<0.01). At exacerbation, there was no difference in VAS symptoms of cough, breathlessness, sputum production or purulence between asthma or COPD at the time of exacerbation nor peripheral blood counts. VAS symptoms and peripheral blood inflammatory mediators at exacerbation did not demonstrate a correlation.

## Characteristics according to D0 treatment

Levels of symptoms and peripheral blood inflammatory mediators compared between participants prescribed both SCS and antibiotic, SCS alone, antibiotics alone, or neither treatment for the exacerbation at D0 are summarised in **Table 2**. Participants who received both SCS and antibiotics, in addition to those who received no treatment had a higher peripheral blood neutrophil count compared to participants who received SCS alone (post-hoc Kruskal Dunn test p<0.01 and p = 0.01 respectively). Eosinophilic inflammation was highest in those prescribed SCS alone as indicated by a higher peripheral blood eosinophil count and percentage (post-hoc Kruskal Dunn test p < 0.01). Participants who received antibiotics had higher D0 VAS sputum symptoms (see **Table 2**).

## Predicting physician SCS prescription decision

SCS were prescribed in 64 (79%) participants. **Table 3** summarises characteristics of participants given SCS or not (for which p < 0.1). The remaining characteristics are summarised in

**Table 2. Comparison of characteristics of participants given SCS only versus participants given antibiotics only versus participants given both SCS and antibiotics versus participants given neither SCS nor antibiotics.**

| Characteristic | D0 Antibiotics and SCS Treatment | | | | P-value |
| --- | --- | --- | --- | --- | --- |
| | **Both** | **SCS Only** | **Antibiotics Only** | **Neither** | |
| | **(n = 31)** | **(n = 33)** | **(n = 6)** | **(n = 11)** | |
| Male n (%) | 11 (35) | 15 (45) | 3 (50) | 5 (45) | 0.82 |
| Current smokers, n (%) | 8 (26) | 8 (24) | 1 (17) | 4 (36) | 0.87 |
| Ex-smokers, n (%) | 18 (58) | 12 (36) | 2 (33) | 2 (18) | 0.29 |
| Never smoker, n (%) | 5 (16) | 13 (39) | 3 (50) | 5 (45) | 0.24 |
| Asthma, n (%) | 20 (65) | 26 (79) | 5 (83) | 8 (73) | 0.91 |
| COPD, n (%) | 11 (35) | 7 (21) | 1 (17) | 3 (27) | 0.69 |
| Number taking ICS, n (%) | 13 (42) | 18 (55) | 4 (67) | 5 (45) | 0.61 |
| Increased wheeze at exacerbation, n (%) | 29 (94) | 29 (88) | 4 (67) | 8 (73) | 0.16 |
| Age (years)* | 55 (39–69) | 45 (28–67) | 54 (46–76) | 49 (34–58) | 0.23 |
| Pack year history* | 14 (2–25) | 3 (0–18) | 0 (0–2) | 1 (0–22) | 0.22 |
| Number of hospital admissions in previous 12 months* | 0 (0–2) | 0 (0–1) | 0 (0–1) | 0 (0–5) | 0.90 |
| Oxygen saturation (%)˜ | 94 (90–98) | 95 (86–100) | 96 (91–100) | 96 (93–100) | 0.36 |
| VAS cough (mm)* | 55 (30–72) | 36 (16–70) | 62 (50–78) | 30 (53–67) | 0.29 |
| VAS breathlessness (mm)* | 76 (50–87) | 65 (26–84) | 70 (61–89) | 70 (0–88) | 0.78 |
| VAS sputum production (mm)* | 28 (8–53) | 13 (0–25) | 31 (24–75) | 2 (0–23) | 0.01 |
| VAS sputum purulence (mm)* | 32 (5–61) | 2 (0–21) | 17 (0–45) | 1 (0–20) | 0.04 |
| Leucocytes (x$10^9$cells/L)* | 11.0 (9.2–14.1) | 9.7 (8.1–12.0) | 11.9 (10.6–14.0) | 11.9 (10.7–13.9) | 0.16 |
| Neutrophils (x$10^9$cells/L)* | 8.9 (6.0–11.0) | 6.0 (4.5–8.2) | 8.1 (6.6–9.0) | 8.1 (7.7–11.1) | 0.03 |
| Neutrophil percentage* | 78.1 (68.6–87.8) | 65.4 (55.4–75.1) | 73.6 (62.6–77.7) | 75.0 (66.5–83.6) | 0.01 |
| Eosinophils (x$10^9$cells/L)* | 0.1 (0.0–0.3) | 0.3 (0.2–0.5) | 0.0 (0.0–0.1) | 0.1 (0.1–0.1) | 0.01 |
| Eosinophil percentage* | 1.2 (0.2–3.1) | 3.0 (1.6–5.5) | 0.4 (0.1–1.1) | 0.9 (0.4–1.1) | 0.00 |
| CRP (mg/L)* | 19.3 (4.6–42.8) | 5.4 (1.0–16.6) | 13.4 (1.7–19.8) | 3.4 (1.3–17.9) | 0.06 |

*Definition of abbreviations*: ICS = inhaled corticosteroids; CRP = C-reactive protein.

*Measures of central tendency and spread*: variables marked with the * symbol are presented as median (IQR); variables marked with the ˜ symbol are presented as mean (range). Comparisons between treatment groups are made using chi-squared, kruskal-wallis or one-way anova test as appropriate. P-values are reported to 2dp unless the p-value is less than 0.001, in which case it is reported as <0.001.

**S1 Table in S1 Appendix**. Participants given SCS had a higher peripheral blood eosinophil percentage compared to participants not given SCS (p < 0.01). Indicators of respiratory disease severity (blood pCO2 and oxygen saturation) were worse in participants given SCS and participants were more likely to report wheeze. Random forest models were developed to determine which combination of biological and symptom variables are the best predictors of the physician decision to prescribe SCS (see **S2 Table in S1 Appendix**). The variables from the random forest multivariate model showed that the presence of increased wheeze at exacerbation together with blood eosinophil percentage were the best. A random forest model using just these two variables achieved an area under receiver operating characteristic curve (AUC) of 0.69 for predicting SCS prescription (see **S3 Table in S1 Appendix** for AUCs of the different models).

## Predicting a treatment failure

In total, there were 43 (53%) treatment failures. Participants who failed treatment were likely to have had a history of exacerbations, a higher symptom burden of dyspnoea, sputum production and purulence at exacerbation and were less likely to be taking inhaled COPD treatment.

**Table 3. Comparison of D0 exacerbation characteristics of patients given SCS versus patients not given SCS for which p < 0.1 (see S1 Appendix for all other characteristics evaluated).**

| Characteristic | Exacerbation | | |
|---|---|---|---|
| | n = 81 Events | | |
| | SCS | No SCS | *P*-value |
| | n = 64 | n = 17 | |
| LAMA, n (%) | 13 (20) | 10 (59) | 0.00 |
| PPI, n (%) | 4 (6) | 5 (29) | 0.01 |
| Eosinophils %* | 1.9 (0.4–4.8) | 0.8 (0.2–1.1) | 0.01 |
| Increased wheeze at exacerbation, n (%) | 58 (91) | 12 (71) | 0.03 |
| Blood pCO2 (kPa)* | 5.7 (5.1–6.1) | 4.8 (4.5–5.7) | 0.05 |
| Total number of ITU admissions in previous 12 months* | 0 (0–0) | 0 (0–1) | 0.05 |
| SCS in week prior to exacerbation admission, n (%) | 33 (52) | 13 (76) | 0.07 |
| Leucocytes (x10$^9$cells/L)* | 10.2 (8.8–12.8) | 11.9 (10.6–14.0) | 0.08 |
| Oxygen saturation (%)˜ | 95 (86–100) | 96 (91–100) | 0.09 |

*Definition of abbreviations*: LAMA = Long-acting muscarinic antagonist; PPI = proton pump inhibitor; ITU = intensive therapy unit.

*Measures of central tendency and spread*: variables marked with the * symbol are presented as median (IQR); variables marked with the ˜ symbol are presented as mean (range). Comparisons between SCS and no SCS group are made using chi-squared, wilcox test or t-test as appropriate. P-values are reported to 2dp unless the p-value is less than 0.001, in which case it is reported as <0.001.

Participants prescribed SCS were also more likely to have a treatment failure. **Table 4** summarises the characteristics of participants who did and did not have a treatment failure for which p < 0.1. The full comparison of characteristics of these participants is summarised in **S4 Table in S1 Appendix**. **Table 5** shows the random forest variable importance scores for the multivariate model containing all variables which passed the univariate p<0.1 filter. The best model used 11 variables and achieved an AUC of 0.81 (see **S5 Table in S1 Appendix** for AUCs of the different random forest models using different combinations and total numbers of variables) for predicting treatment failure. VAS breathlessness and VAS sputum purulence at exacerbation contribute most to the predictive performance. A random forest model using just these two variables predicted treatment failure with AUC 0.68.

## Discussion

In this study, we have described the characteristics of patients with asthma and COPD who present to the ED with an exacerbation. We have shown that treatment failure with SCS and/or antibiotic therapy occurs in approximately 50% and that symptoms of breathlessness and sputum purulence appear be good predictors of a treatment failure following an exacerbation of airways disease. In addition to this, the clinician decision to prescribe SCS is related to several factors. Increased wheeze at exacerbation and peripheral blood eosinophil percentage together predict SCS prescription at the onset of an exacerbation of asthma and/or COPD, although it is possible that increased wheeze at exacerbation is a confounder for the relationship between peripheral blood eosinophil percentage and SCS prescription.

In this study we have used random forest analyses to identify factors associated with exacerbation treatment failure. We showed that patients who went on to fail treatment had more pronounced breathlessness, sputum production and sputum purulence at exacerbation, indicative of symptom burden. Exacerbations are symptom-defined events [1] and our findings illustrate this. Additionally our results potentially allow for quantitative qualification of the symptoms

**Table 4. Comparison of D0 exacerbation characteristics of patients who failed treatment with those who succeeded treatment which p < 0.1 (see S1 Appendix for all other characteristics evaluated).**

| Characteristic | Exacerbation | | |
| --- | --- | --- | --- |
| | n = 81 Events | | |
| | Treatment Failure, n = 43 | Treatment Success, n = 38 | *P*-value |
| VAS sputum production (mm)* | 28 (10–56) | 8 (0–25) | 0.00 |
| VAS dyspnoea (mm)* | 78 (60–89) | 55 (4–78) | 0.00 |
| Number of exacerbations associated with increased sputum production, n (%) | 28 (65) | 13 (34) | 0.01 |
| Number of unscheduled primary care and emergency department visits in previous 12 months* | 2 (1–5) | 1 (0–3) | 0.01 |
| Number taking ICS, n (%) | 15 (35) | 25 (66) | 0.01 |
| VAS sputum purulence (mm)* | 40 (0–68) | 3 (0–15) | 0.01 |
| Number of exacerbations associated with increased sputum purulence, n (%) | 22 (51) | 10 (26) | 0.02 |
| SCS at exacerbation, n (%) | 38 (88) | 26 (68) | 0.03 |
| Oxygen saturation, n (%)˜ | 94 (86–100) | 96 (91–100) | 0.03 |
| SABA, n (%) | 32 (74) | 35 (92) | 0.04 |
| LAMA, n (%) | 8 (19) | 15 (39) | 0.04 |
| PPI, n (%) | 2 (5) | 7 (18) | 0.05 |
| VAS cough (mm)˜ | 54 (0–99) | 43 (0–100) | 0.08 |

*Definition of abbreviations*: ICS = inhaled corticosteroids; SABA = short-acting beta agonist; LAMA = Long-acting muscarinic antagonist; PPI = proton pump inhibitor.
*Measures of central tendency and spread*: variables marked with the * symbol are presented as median (IQR); variables marked with the ˜ symbol are presented as mean (range). Comparisons between treatment failure and treatment success are made using chi-squared, wilcox test or t-test as appropriate. P-values are reported to 2dp unless the p-value is less than 0.001, in which case it is reported as <0.001.

measured at the onset of the event that are associated with exacerbations that fail to respond to management at the onset. Furthermore, patients who were on inhaled therapy (corticosteroids, short-acting beta agonists and/or long-acting muscarinic antagonists) were less likely to have a treatment failure following an exacerbation, reiterating that undertreated COPD is associated with a higher chance of a patient requiring further healthcare utilisation [21]. We also showed that the use of PPI appeared to be protective of a treatment failure. This was interesting as gastro-oesphageal reflux disease has previously been shown to be an independent predictor of exacerbations [22, 23].

Our random forest analysis to predict treatment failure showed that a random forest model using a combination of breathlessness and sputum purulence exacerbation symptom VAS scores could effectively predict treatment failure for patients irrespective of the type of exacerbation treatment patients were given. Our finding that breathlessness at exacerbation is a useful predictor of treatment failure compliments findings of other studies. Breathlessness is already known to be a predictor of 5 year mortality for COPD patients [24] whilst the Medical Research Council Dyspnoea Scale is a predictor of both hospital mortality and 28-day exacerbation readmission [25]. Similarly, dyspnoea has been shown to be associated with exacerbation relapse risk in patients with acute exacerbations of COPD [26]. Our finding that sputum purulence is a predictor of exacerbation treatment failure is interesting in the context of GOLD and NICE guidance that antibiotics should be given at exacerbation in patients with increased sputum purulence since this may reduce exacerbation relapse and treatment failure [1, 27]. The relatively high prescription of antibiotics for asthma exacerbations in our study is not an uncommon occurrence, as has previously been shown [28]. This is likely due to patients not being seen by specialists in ED and reflects clinical practice, further emphasising the importance of developing tools to assist non-specialists in ED. In our study it was observed that patients receiving antibiotics at exacerbation did indeed have significantly greater sputum

**Table 5. Random forest importance scores of variables in multivariate treatment failure prediction models for all participants.**

| Name of Successive Variable | Importance Score of Successive Variable |
|---|---|
| VAS breathlessness | 11.2 |
| VAS sputum purulence | 8.3 |
| LAMA | 6.5 |
| Number of unscheduled primary care and emergency department visits in previous 12 months | 5.6 |
| ICS use | 4.9 |
| VAS sputum production | 4.2 |
| PPI use | 4.2 |
| EuroQol mobility | 4.1 |
| Oxygen saturation | 3.8 |
| SABA use | 3.3 |
| SCS at exacerbation | 2.7 |
| Exacerbation associated with increased sputum purulence | 1.1 |
| Exacerbation associated with increased sputum production | 1.0 |
| EuroQol self care | 0.8 |
| Gender | 0.6 |
| EuroQol usual activity | -0.5 |
| VAS cough | -1.0 |

*Definition of abbreviations*: LAMA = Long-acting muscarinic antagonist; ICS = inhaled corticosteroids; PPI = proton pump inhibitor; SABA = short-acting beta agonist.

Models shown range from those using just a single variable up to those with the full subset which passed the univariate analysis filter of p < 0.1. Variable importance scores were calculated as outlined in S1 Appendix Statistical Methods. To illustrate the interpretation of the variable importance scores, consider the example of VAS breathlessness. The VAS breathlessness random forest importance score of 11.2 indicates that classification accuracy would drop by 11.2% if VAS breathlessness is omitted from the classification model.

purulence compared to patients not receiving antibiotics. Therefore, it is unlikely that treatment failures in our study resulted from inappropriate or non- prescription of antibiotics at exacerbation. Rather, our finding that higher dyspnoea and sputum purulence are predictive of treatment failure may simply reflect that patients with greater disease burden and more severe exacerbations in terms of symptomatic presentation are more likely to fail treatment. It may also be the case that the patients with the highest symptom load always have a high symptom load, so the threshold to declare this is reached more frequently and treatment may not lower their symptoms enough for them to drop below their threshold. Our conclusion that treatment failure can be predicted based on exacerbation symptomatological presentation reaffirms the need for clinicians to pay close attention to patient symptom presentation at exacerbation. Furthermore, the ease of measuring patient symptoms through the VAS score would make treatment failure prediction models like those in our study easy to implement in clinical practice.

It is noteworthy that a high symptom burden for VAS sputum production and sputum purulence in addition to CRP were characteristic of patients given antibiotics, although CRP did not reach statistical significance. It has been shown that using CRP as a biomarker to direct antibiotic treatment of severe COPD exacerbations is effective in achieving good clinical treatment outcomes as well as reducing antibiotic prescription [29]. The latter is especially important given the concerns associated with excessive antibiotic use for treating exacerbations

promoting antibiotic resistance [30]. The only biological exacerbation characteristic which differed between patients given SCS and patients not given SCS was the degree of eosinophilic inflammation. Blood results would not have been available for clinicians prior to treatment initiation. This is an interesting finding, as it suggests that there is something about eosinophilic inflammation that drives a clinician to prescribe SCS. In investigating this further, our univariate analysis, showed that prescription of a LAMA and PPI, was associated with reduced SCS prescription, whilst an elevated $pCO_2$, and the presence of wheeze was associated with SCS prescription. Random forest models revealed that blood eosinophil percentage and increased wheeze at exacerbation were predictors of SCS prescription. Eosinophilic inflammation is related to increased luminal airway oedema [31] and in asthma post-mortem narrowed airways [32], which could suggest that eosinophilic inflammation is related to wheeze and a greater degree of clinical severity of the presentation. In the context of our study, this may explain the greater predominance of wheeze at exacerbation as well as eosinophilic inflammation for those patients given SCS. Studies have shown that patients with a high peripheral blood eosinophil count respond better to SCS than patients with a low blood eosinophil count [10] and our finding suggests the importance of measuring eosinophils in all exacerbations of airways disease.

Our study has some limitations. Firstly, combination of asthma and COPD may make interpretation difficult. However, as is standard practice, often there is no access to lung function in ED from primary care. In addition, where monitoring is now impacted by the absence of spirometry in the community setting due to COVID, it is prudent to make the assumption that a clear-cut diagnosis of asthma or COPD is not always available. The combination of asthma and COPD can thus be seen as an advantage rather than a limitation. A second potential limitation is that we did not perform SCS prescription and treatment failure analyses for asthma compared to COPD patients. This is because out of the 81 patients, only 22 had a primary diagnosis of COPD, which would be too small a sample size for separate analysis. However, whether the primary diagnosis was asthma or COPD is a variable considered in our SCS prescription and treatment failure prediction analysis and this variable is not selected in the best random forest models. Hence, we conclude that whether the diagnosis is asthma versus COPD is not related to SCS prescription and treatment failure in our study cohort. We note that a third limitation of our study is the lack of external validation. However, our leave-one-out cross-validation strategy already enabled validation of different models to be performed in test subjects. In the context of our small sample size, this is superior compared to splitting the dataset into one training and one testing portion which would waste data otherwise available for training [20]. Furthermore, our choice of random forest as the supervised learning approach was strategic to reduce the potential of over-fitting [19]. We envisage that in additional future clinical validation studies it will be more reliable to use the variables in their original form rather than in a transformed form for input into the multivariate classifier being assessed in a validation study. Therefore, in our univariate analysis we compared different types of p-values including chi-squared, t-test and Wilcox test. The mixture of continuous, discrete, binary and categorical variables in the study made comparing at least two different types of p-values unavoidable. For our univariate analysis we did not correct for multiple comparisons since the univariate analysis was purely exploratory in nature. Nevertheless, we note that the data-driven, assumption-free machine learning approach in our study is a key strength. It should be noted that when narrowing down the size of variable subsets in final multivariate models for treatment failure prediction, we deliberately used the random forest variable importance score and feature elimination procedure rather than relying on potentially more easily interpretable univariate p-values or AUCs of single variable models to indicate variable importance. Ordering variables according to the corresponding p values from the initial

univariate analysis would be mathematically inaccurate since: 1) the p-value would only indicate univariate importance of a variable which may be very different from the importance of the variable in a multivariate context [20]; 2) the p-values relate to the hypothesis test for a difference between the treatment failure and treatment success groups rather than being a direct measure of the ability of the variable to classify [20] a patient as being in the treatment failure versus treatment success group. In addition, it is mathematically best to use a variable ranking system directly related to the particular classifier model being used since this provides more information regarding the generalisability of the variable's importance for classification using the relevant model [20].

To the best of our knowledge, our study is the first to assess a combination of VAS-based symptoms, biomarkers, clinical characteristics and demographics as predictors of treatment failure in a mixed cohort of hospitalised asthma and COPD patients.

## Conclusion

We have shown that over half of all asthma and COPD patients admitted to ED for their exacerbation require additional major medication and/or readmission to the emergency department within 30 days of their exacerbation. We have also shown that prescription of SCS at exacerbation by physicians may be related to presentation of wheeze and may be driven by eosinophilic inflammation. Furthermore, breathlessness and sputum purulence are useful for predicting whether additional major medication and/or readmission to the emergency department within 30 days of exacerbation will occur.

## Supporting information

**S1 Appendix. Predicting treatment outcomes following an exacerbation of airways disease appendix.**
(DOCX)

**S1 Dataset.**
(CSV)

## Acknowledgments

The authors thank all the research volunteers who participated in the study.

## Author Contributions

**Conceptualization:** Mona Bafadhel, Richard E. K. Russell.

**Data curation:** Sally Beer, Richard Pullinger.

**Formal analysis:** Andreas Halner.

**Investigation:** Mona Bafadhel, Richard E. K. Russell.

**Methodology:** Andreas Halner, Mona Bafadhel, Richard E. K. Russell.

**Supervision:** Mona Bafadhel, Richard E. K. Russell.

**Validation:** Andreas Halner.

**Writing – original draft:** Andreas Halner.

**Writing – review & editing:** Mona Bafadhel, Richard E. K. Russell.

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
