## [Editor Report · Decision Letter 0]

2 Feb 2021

PONE-D-20-36305

Predicting Treatment Outcomes Following an Exacerbation of Airways Disease

PLOS ONE

Dear Dr. Halner,

Thank you for submitting your manuscript to PLOS ONE. After careful consideration, we feel that it has merit but does not fully meet PLOS ONE’s publication criteria as it currently stands. Therefore, we invite you to submit a revised version of the manuscript that addresses the points raised during the review process.

The goal of the paper is to build a parsimonious statistical predictive model for treatment failure of COPD and asthma exacerbation. The topic is important and in general the approach is acceptable. However, several issues must be fixed before the paper can be sent out for full review.

Some page numbers are shown and some not – it makes reading of the hard copy difficult.Comparing p-values obtained by different methods is cumbersome. Why authors don’t use the traditional t-test, perhaps after normalizing transformation, such as taking log?It is unclear what P-value in Table 2 comes from. What is compared? I suggest showing p-values in scientific format, like 1.2x10^-4^, instead <0.01. Such p-value may help in comparing the strength of predictors.The notation β=1.72 on page 11 comes from nowhere. The coefficient?I do not see the ref for NagelKerke’s  R^2^. I think using AUC, or the C-statistic, is a more appropriate for this study as the measure of correct classification.Why R^2^ is blank for univariate models in Table 4?The table results, such as presented in Table 2, are difficult to read. I suggest boxplots instead. This comment applies to all other results including the ROC curve.The authors claim that the data are fully available. However neither Excel file nor URL is provided where the raw data can be downloaded from.Random forest results on importance of predictors in Table 6 are questionable because of lack of interpretation. How to interpret the scores and what they mean? Instead, the variables may be ordered with respect to the p-value or better off with respect to increasing values of AUC from expanding logistic regression. The AUC has clear interpretation as % failure prediction.I don’t see a good reason to include variables with p-value p>=0.03 in the final predictive model presented in Table 5. Instead I suggest computing the C-statistic (AUC) for expanding set of variables. Then the contribution of each predictor could be easily assessed.The choice of the final model is not justified well. I suggest BIC or AIC criterion.

We look forward to receiving your revised manuscript.

Kind regards,

Eugene Demidenko, Ph.D.

Academic Editor

PLOS ONE

2.We note that the grant information you provided in the ‘Funding Information’ and ‘Financial Disclosure’ sections do not match.

3.Thank you for stating the following in the Competing Interests section:

"I have read the journal's policy and the authors of this manuscript have the following competing interests:

Mona Bafadhel reports outside the submitted work research grant reports from AZ; honoraria from AZ, Chiesi, and GlaxoSmithKline; and is on the scientific advisory board for AlbusHealth® and ProAxsis®. Richard Russell has received honoraria from AZ, GSK, Boheringer Ingelheim, Chiesi, Cipla and is on the advisory board for AlbusHealth®, has received research funding from Circassia UK and his work is supported by the Oxford NIHR Biomedical Research Centre.

The remaining authors have declared that no competing interests exist."

---

## [Author Response · Author response to Decision Letter 0]

4 Mar 2021

RE: Predicting Treatment Outcomes Following an Exacerbation of Airways Disease

REF: PONE-D-20-36305

To the Editor, Dr Eugene Demidenko, 

Thank you very much for your consideration of our manuscript. We have taken the time to thoroughly address the reviewer comments through making the suggested edits or providing a justification of our approach. As requested, we have uploaded a tracked changes version named ‘Revised Manuscript with Track Changes PONE-D-20-36305’ (as well as ‘Revised Supplement with Track Changes - PONE-D-20_36305’) and an unmarked version named ‘Manuscript - PONE-D-20-36305_Clean’ (as well as ‘Supplement - PONE-D-20_36305_Clean’).

Please see our responses below: 

Comment 1: “Some page numbers are shown and some not – it makes reading of the hard copy difficult.”

Response 1: Thank you for pointing this out. This has been now addressed.

Comment 2: “Comparing p-values obtained by different methods is cumbersome. Why authors don’t use the traditional t-test, perhaps after normalizing transformation, such as taking log?”

Response 2: Thank you. We have intentionally not applied data transformations since we envisage that in future clinical validation studies it will be more reliable to use the variables in their original form rather than in a transformed form for input into the multivariate classifier being assessed in a validation study. This is important since for future-obtained data one cannot be sure of the distribution (e.g., whether normal or not) and since there is a mixture of continuous, discrete, binary or categorical variables comparing at least two types of p-values (i.e., chi-squared and t-test or Wilcox test) is unavoidable. We have discussed this now further in the main body of the manuscript (see ‘Revised Manuscript with Track Changes PONE-D-20-36305’ ‘Statistical Analyses’ section on pg.6 lines 11-12 and ‘Discussion’ on pg.19 lines 18-23). 

Comment 3: “It is unclear what P-value in Table 2 comes from. What is compared? I suggest showing p-values in scientific format, like 1.2x10-4, instead <0.01. Such p-value may help in comparing the strength of predictors.”

Response 3: Thank you this has now been addressed and we have now replaced < 0.01 with more precise p-values via the use of standard form, as suggested by the reviewers. We have added detail to the title of Table 2 to clarify the nature of the comparisons between treatment groups. For the sake of consistency, we have now also used standard form for p-values in all other tables in the ‘Revised Manuscript with Track Changes PONE-D-20-36305’. 

Comment 4: “The notation β=1.72 on page 11 comes from nowhere. The coefficient?”

Response 4: Thank you this has now been addressed. We have clarified this by writing “β coefficient” rather than just “β” (see ‘Revised Manuscript with Track Changes PONE-D-20-36305’ pg.12 line 10).

Comment 5: “I do not see the ref for NagelKerke’s R2. I think using AUC, or the C-statistic, is a more appropriate for this study as the measure of correct classification.”

Response 5: Thank you this has now been addressed. For each model in Table 4, instead of the NagelKerke’s R2 we have now reported the AUC in addition to Akaike Information Criterion (AIC), with one exception for the univariate model which uses the binary variable wheeze. In the latter case, the AIC but not the AUC is reported. We have outlined the reporting of AUC and AIC, as well as the lack of AUC reporting for the univariate binary predictor case (with references), in the ‘Revised Manuscript with Track Changes PONE-D-20-36305’ (pg.6 line 23 and pg.7 lines 1-2; pg.12 line 12) and ‘Revised Supplement with Track Changes - PONE-D-20_36305' document (see pg.2 lines 10-16).

Comment 6: “Why R2 is blank for univariate models in Table 4?”

Response 6: Thank you. As per comment 5, the R2 in Table 4 has been replaced with AIC and AUC.

Comment 7: “The table results, such as presented in Table 2, are difficult to read. I suggest boxplots instead. This comment applies to all other results including the ROC curve.”

Response 7: Thank you. We have clarified table 2 for readability. We have not included box-plots as per reviewers suggestion as this would require 56 individual plots. If the editorial discretion is to replace table 2 with box-plots we can upload this. 

Comment 8: “The authors claim that the data are fully available. However, neither Excel file nor URL is provided where the raw data can be downloaded from.”

Response 8: The data is fully available following written request. This is clarified in the manuscript acknowledgements (see ‘Revised Manuscript with Track Changes PONE-D-20-36305’ ‘Acknowledgements’ section on pg.22).

Comment 9: “Random forest results on importance of predictors in Table 6 are questionable because of lack of interpretation. How to interpret the scores and what they mean? Instead, the variables may be ordered with respect to the p-value or better off with respect to increasing values of AUC from expanding logistic regression. The AUC has clear interpretation as % failure prediction.”

Response 9: Thank you. In our manuscript we have ordered the random forest variable importance scores to show the reader the hierarchy of importance of the variables as contributing to the predictive ability of the multivariate random forest model. For example, presenting a hierarchy of the p value would indicate the importance of the variable in univariate analysis only; whilst the AUCs for random forest models using different combinations of variables are provided in the supplement S1 Appendix Table 3. To add clarity, we have added in the footnote of Table 6 what the importance scores translate to. We have used the example that the VAS breathlessness random forest importance score of 11.2 indicates that classification accuracy would drop by 11.2% if this was omitted. 

Comment 10: “I don’t see a good reason to include variables with p-value p>=0.03 in the final predictive model presented in Table 5. Instead, I suggest computing the C-statistic (AUC) for expanding set of variables. Then the contribution of each predictor could be easily assessed.”

Response 10: Thank you. As described in the supplement S1 Appendix we have included all variables with univariate p values < 0.1 in the initial multivariate model and the selected variables (17 variables met the p < 0.1 univariate filter in our treatment failure prediction analysis) were ranked according to random forest variable importance scoring. We have clarified this further as per response 9. We deliberately used an initial p < 0.1 univariate filter rather than a more limiting threshold (e.g., p <0.05 or the p < 0.03 suggestion made by the reviewer), because the initial filter step serves to include as many variables as possible for consideration in the multivariate model but to remove variables likely to represent noise rather than being truly informative. We have added further details in the statistical appendix (see ‘Revised Supplement with Track Changes - PONE-D-20_36305' pg.2 lines 28-33 and pg.3 lines 1-2) as to why this method was chosen and discussed further other methods (e.g., the C-Statistic as per 

Comment 10 and p-values as per Comment 9) and their limitations in the main manuscript (see ‘Revised Manuscript with Track Changes PONE-D-20-36305’ pg.20 lines 2-15). 

Comment 11: “The choice of the final model is not justified well. I suggest BIC or AIC criterion.”

Response 11: Thank you. Although these criterions could be used, in our model, neither BIC nor AIC can be applied in the context of an ensemble decision tree-based classifier such as the random forest. Both the number of parameters estimated by a classifier and the maximum value of the likelihood function for a model must be known in order to compute BIC or AIC. The primary reason why we used the random forest classifier for predicting treatment failure in our study rather than using a linear parametric model such as the logistic regression is that random forests can be effective in preventing overfitting, even in a low sample size high dimensional context. We have further discussed in the statistical appendix why we selected this model and why AIC/BIC criterion cannot be applied (see ‘Revised Supplement with Track Changes - PONE-D-20_36305' pg.3 lines 11-31 and pg.11 references 1 and 5). 

Kind regards,

Mr Andreas Halner & Professor Mona Bafadhel

on behalf of the co-authors

---

## [Decision Letter · Decision Letter 1]

11 Mar 2021

PONE-D-20-36305R1

Predicting Treatment Outcomes Following an Exacerbation of Airways Disease

PLOS ONE

Dear Dr. Halner,

Thank you for submitting your manuscript to PLOS ONE. After careful consideration, we feel that it has merit but does not fully meet PLOS ONE’s publication criteria as it currently stands. Therefore, we invite you to submit a revised version of the manuscript that addresses the points raised during the review process.

The reviews of your manuscript are on the negative side: the first reviewer suggested revision and the second reviewer suggested rejection. Nevertheless, I give you a chance to resubmit. Addressing their concerns in a point-by-point fashion is imperative. Lack of convincing response will result in the subsequent rejection. Especially important is the critique of the second reviewer who made very important objections from the medical perspective. Feel free to withdraw the paper if you think that such response would be difficult provide.

We look forward to receiving your revised manuscript.

Kind regards,

Eugene Demidenko, Ph.D.

Academic Editor

PLOS ONE

Reviewers' comments:

Reviewer's Responses to Questions

**Comments to the Author**

1. If the authors have adequately addressed your comments raised in a previous round of review and you feel that this manuscript is now acceptable for publication, you may indicate that here to bypass the “Comments to the Author” section, enter your conflict of interest statement in the “Confidential to Editor” section, and submit your "Accept" recommendation.

Reviewer #1: (No Response)

Reviewer #2: (No Response)

2. Is the manuscript technically sound, and do the data support the conclusions?

Reviewer #1: Partly

Reviewer #2: No

3. Has the statistical analysis been performed appropriately and rigorously? 

Reviewer #1: No

Reviewer #2: I Don't Know

4. Have the authors made all data underlying the findings in their manuscript fully available?

Reviewer #1: No

Reviewer #2: Yes

5. Is the manuscript presented in an intelligible fashion and written in standard English?

Reviewer #1: Yes

Reviewer #2: Yes

6. Review Comments to the Author

Reviewer #1: Overall, I don't think the methodology has been sufficiently thought through. To give you the benefit of the doubt, the wide mix of analyses and methods mean that you have not been able to thoroughly describe them. I think your aims are a little unclear - are you just looking for factors associated with SCS prescription and treatment failure, or are you trying to predict them? If the latter, the methodology is inappropriate because you are not reporting or testing the performance in a valid way, and the models are not being developed well. If the former, reporting the AIC and AUC seems unnecessary, as does comparing the results across different numbers of variables included.

I think you need to refine your analysis question, and remove some of the analyses.

• Abstract: SCS acronym used in abstract without introduction

• Introduction: please define the ‘standard initial treatment’

• Methods:

o Were there any participants with asthma-COPD overlap syndrome? This is not mentioned at all.

o It is unclear to me why the methods were not consistent between the two main analyses: SCS prescription and treatment failure. You justify the use of the random forest for the second analysis, but it seems like it would have been appropriate for the first analysis for the same reason.

o Insufficient detail is provided on the construction of the multivariate logistic regression models. Did you try multiple combinations of between two and four variables and then select the best performing? Why was a stepwise approach not used?

o By only allowing the random forest to choose from variable significant in the univariate logistic regression, potential interactions between variables are very limited. The ability to calculate such interactions is a key strength of tree-based algorithms.

o Which implementation of the random forest algorithm was used? What were the hyperparameters?

o Did you look at the analyses stratified by diagnosis? It’s reasonable to assume it might differ between asthma and COPD.

• Results:

o Please change ‘asthmatics’ to ‘people with asthma’

o No p-values appeared to be in bold in any of the tables.

o I disagree with the previous reviewer that presenting the p-values in standard form is better, I think your previous approach was better.

o Table 2 just says LAMA, but table 6 says lama use. Be consistent.

o Table 2 – can you group the characteristics by type (continuous, binary etc) for ease of reading?

o Please define acronyms used in tables in the notes underneath them

• Discussion:

o You should not be defining acronym for the first time in the discussions when it has been used throughout, such as PPI.

o Grammatical error: “so the threshold to declare this, is reached more frequently”

o Overly strong assumption of causal relationship between eosinophilic inflammation and SCS prescribing. It seems very possible that this is confounding by wheeze, or similar.

Reviewer #2: The authors at John Radcliffe Hospital conducted this prospective observational study aiming to develop predictive models for exacerbation treatment outcome for patients with asthma and COPD exacerbation. They included 81 patients in their final analysis (59 asthma patients and 22 COPD patients). They first did a univariate analysis comparing the characteristics of patients who did or didn't receive systemic corticosteroids followed by multivariable logistic regression model to predict biological and historic predictors of physicians' decision to prescribe corticosteroids. After that they performed random forest models to find the predictors of treatment failure. I have three major concerns about this manuscript behind my recommendation to reject this manuscript:

First, I don't understand what is the utility of finding the predictors whether or not to prescribe systemic corticosteroids for asthma and COPD exacerbation? The authors cited in the introduction that there is "a paucity of data which demonstrate inconsistent benefits for SCS and/or antibiotics for treating exacerbations of COPD and asthma in which treatment is received in hospital". The use of systemic steroids in asthma and COPD exacerbation is standard of care and has grade A evidence in the guidelines. There is grade A evidence (quote from GOLD guidelines) that "SCS use in COPD exacerbation can improve FEV1, oxygenation and shorten recovery time and hospitalization duration". As a pulmonologist, I believe if the diagnosis of asthma or COPD exacerbation is confirmed or highly suspected, the patient should be given SCS unless there is a contraindication or a reason for the treating physician not to give it. In this study, 21% of the patients didn't get corticosteroids. Additionally, 42% of the asthma exacerbation patients received antibiotics which isn't standard of care for asthma exacerbation patients which make me suspect there was some clinical suspicion for infection/pneumonia in these patients.

Second, I find it methodologically troublesome to combine COPD and asthma patients in one basket and try to extrapolate prediction models from this combined group. While both asthma and COPD are obstructive lung diseases, there are big differences in the pathophysiology and patient populations. another minor point here, it's not clear how the diagnosis of COPD and asthma was made. did the patients have to have a pulmonary function test prior to the Ed visit or did they have to have radiologic evidence of emphysema for COPD at least?

Third, generating reliable prediction models requires larger sample size. As a clinician, if I read this article as a reader, I won't be able to draw reliable conclusions or use the prediction model presented in this manuscript as it was derived from 81 patients with no validation.

7. PLOS authors have the option to publish the peer review history of their article (what does this mean?). If published, this will include your full peer review and any attached files.

Reviewer #1: No

Reviewer #2: No

---

## [Author Response · Author response to Decision Letter 1]

24 Mar 2021

To the Editor, Dr Eugene Demidenko, 

Thank you very much for the opportunity you have provided us to respond to the reviewers. We have taken the time to make the changes/additions and explanations suggested by the reviewers. The main changes are 1) we have now consistently used just one type of algorithm, the random forest, throughout the manuscript; 2) expanded upon limitations in our Discussion section; 3) expanded upon the relevance of our study in our Discussion section. We have uploaded a tracked changes version and an unmarked version both for the main manuscript and for the supplement.

Please see our responses to the two reviewers below:

Reviewer 1

Overall comments: “Overall, I don't think the methodology has been sufficiently thought through. To give you the benefit of the doubt, the wide mix of analyses and methods mean that you have not been able to thoroughly describe them. I think your aims are a little unclear - are you just looking for factors associated with SCS prescription and treatment failure, or are you trying to predict them? If the latter, the methodology is inappropriate because you are not reporting or testing the performance in a valid way, and the models are not being developed well. If the former, reporting the AIC and AUC seems unnecessary, as does comparing the results across different numbers of variables included. I think you need to refine your analysis question, and remove some of the analyses.” 

Response: Thank you for your reflections and helpful comments. We have addressed them and we believe that the clarity of the work has been improved. This study is aimed at exploring: 1) variables associated with SCS prescription; 2) variables associated with treatment failure; 3) a multivariate model for predicting SCS prescription and predicting treatment failure. In keeping with your comments and for clarity, we have now used only one method for SCS prescription and treatment failure (random forest). We now discuss in the Methods and Discussion section the use of random forest models in our study and their strengths and limitations. We compare random forest models using different variable subset combinations in order to determine which models are both parsimonious and perform with high accuracy as per suggestion by the reviewer. We have removed all logistic regression analyses, including the corresponding AIC/ AUC which was requested by the previous reviewer. 

Comment 1: “Abstract: SCS acronym used in abstract without introduction”.

Response 1: Thank you for spotting. This has now been addressed (page 3: lines 38-39) and used throughout after introduction.

Comment 2: “Introduction: please define the ‘standard initial treatment’”

Response 2: We have now added the clarification “as per NICE guidelines (with SCS and/or antibiotics)” with accompanying references on page 4, lines 62-63.

Comment 3: “Methods: Were there any participants with asthma-COPD overlap syndrome? This is not mentioned at all.”

Response 3: Thank you for your suggestion. We did not include this term since the asthma-COPD overlap syndrome term has been removed from any guidance and in UK clinical practice is not used. 

Comment 4: “Methods: It is unclear to me why the methods were not consistent between the two main analyses: SCS prescription and treatment failure. You justify the use of the random forest for the second analysis, but it seems like it would have been appropriate for the first analysis for the same reason.” 

Response 4: Thank you very much for your helpful comment. We fully agree with you that a random forest is the better model to use in both cases. Throughout all of the manuscript, random forest methodology and results reporting has now been used in place of a logistic regression analysis. 

Comment 5: “Methods: Insufficient detail is provided on the construction of the multivariate logistic regression models. Did you try multiple combinations of between two and four variables and then select the best performing? Why was a stepwise approach not used?” 

Response 5: As per comment 4, we have now employed a random forest model both for SCS prescription and treatment failure prediction. Details of the random forest methodology are provided in the Methods section of the main manuscript and in the S1 Appendix Statistical Analysis (main manuscript page 7, lines 129-130 and page 8, lines 133-135; supplement page 2, lines 19-51).

Comment 6: “Methods: By only allowing the random forest to choose from variable significant in the univariate logistic regression, potential interactions between variables are very limited. The ability to calculate such interactions is a key strength of tree-based algorithms.” 

Response 6: Thank you. We agree that the random forest model with feature elimination subsequently enables the best combination of variables to be used (taking into account interactions between variables) for the final SCS prescription prediction and treatment failure prediction models. 

Comment 7: “Methods: Which implementation of the random forest algorithm was used? What were the hyperparameters?”

Response 7: Thank you. We have added the implementation and model hyperparameter details in the S1 Appendix Statistical Analysis section (page 2, lines 20-23). The number of trees in the random forest models is 1000. The number of variables to be randomly selected at each split in each tree node was equal to the square root of the number of predictors being considered (number of predictors depend on the step of the feature elimination process).

Comment 8: “Methods: Did you look at the analyses stratified by diagnosis? It’s reasonable to assume it might differ between asthma and COPD.” 

Response 8: We intentionally did not perform the predictive analyses stratified by diagnosis. This is for two reasons: 1) Out of the 81 patients, only 22 had a primary diagnosis of COPD. This would be too small a sample size for separate analysis; 2) whether the primary diagnosis was asthma or COPD was a variable considered in our SCS prescription and treatment failure prediction analysis. However, this variable was not selected in the best models, so it can be mathematically seen that this is not related to SCS prescription and treatment failure in our study cohort. This point is also discussed in the paper (page 20, lines 343-358).

Comment 9: “Results: Please change ‘asthmatics’ to ‘people with asthma’” 

Response 9: Thank you – this has now been addressed (page 9, line 146).

Comment 10: “Results: No p-values appeared to be in bold in any of the tables.” 

Response 10: Thank you – this has now been addressed for tables throughout the manuscript. No p-values are in bold and the table footnotes no longer mention that any p values are bolded.

Comment 11: “Results: I disagree with the previous reviewer that presenting the p-values in standard form is better, I think your previous approach was better.” 

Response 11: Thank you – we also preferred the non-standard form. We have now reported p-values to 2dp but in keeping with PLOS ONE submission guidelines, p-values less than 0.001 are reported as p < 0.001. For tables where this applies, we have now indicated this in the table footnotes. We are happy to follow the Journal Editors selection as required.

Comment 12: “Results: Table 2 just says LAMA, but table 6 says lama use. Be consistent.”

Response 12: Thank you for spotting – we have now addressed this and used “LAMA” throughout all tables.

Comment 13: “Results: Table 2 – can you group the characteristics by type (continuous, binary etc) for ease of reading?”

Response 13: We have now addressed this.

Comment 14: “Results: Please define acronyms used in tables in the notes underneath them” 

Response 14: Thank you - we have now addressed this throughout.

Comment 15: “Discussion: You should not be defining acronym for the first time in the discussions when it has been used throughout, such as PPI.” 

Response 15: Thank you – we have now defined PPI in table footnotes in the Results section.

Comment 16: “Discussion: Grammatical error: “so the threshold to declare this, is reached more frequently” 

Response 16: Thank you – we have deleted the comma (see page 19, line 312).

Comment 17: “Discussion: Overly strong assumption of causal relationship between eosinophilic inflammation and SCS prescribing. It seems very possible that this is confounding by wheeze, or similar.”

Response 17: We have now added that “…it is possible that increased wheeze at exacerbation is a confounder for the relationship between peripheral blood eosinophil percentage and SCS prescription.” (see page 17, lines 272-273). In addition, this is discussed in more detail on page 19 line 335 and page 20 lines 336-339.

Reviewer 2

Overall comments: “The authors at John Radcliffe Hospital conducted this prospective observational study aiming to develop predictive models for exacerbation treatment outcome for patients with asthma and COPD exacerbation. They included 81 patients in their final analysis (59 asthma patients and 22 COPD patients). They first did a univariate analysis comparing the characteristics of patients who did or didn't receive systemic corticosteroids followed by multivariable logistic regression model to predict biological and historic predictors of physicians' decision to prescribe corticosteroids. After that they performed random forest models to find the predictors of treatment failure. I have three major concerns about this manuscript behind my recommendation to reject this manuscript:

First, I don't understand what is the utility of finding the predictors whether or not to prescribe systemic corticosteroids for asthma and COPD exacerbation? The authors cited in the introduction that there is "a paucity of data which demonstrate inconsistent benefits for SCS and/or antibiotics for treating exacerbations of COPD and asthma in which treatment is received in hospital". The use of systemic steroids in asthma and COPD exacerbation is standard of care and has grade A evidence in the guidelines. There is grade A evidence (quote from GOLD guidelines) that "SCS use in COPD exacerbation can improve FEV1, oxygenation and shorten recovery time and hospitalization duration". As a pulmonologist, I believe if the diagnosis of asthma or COPD exacerbation is confirmed or highly suspected, the patient should be given SCS unless there is a contraindication or a reason for the treating physician not to give it. In this study, 21% of the patients didn't get corticosteroids. Additionally, 42% of the asthma exacerbation patients received antibiotics which isn't standard of care for asthma exacerbation patients which make me suspect there was some clinical suspicion for infection/pneumonia in these patients. Second, I find it methodologically troublesome to combine COPD and asthma patients in one basket and try to extrapolate prediction models from this combined group. While both asthma and COPD are obstructive lung diseases, there are big differences in the pathophysiology and patient populations. another minor point here, it's not clear how the diagnosis of COPD and asthma was made. did the patients have to have a pulmonary function test prior to the Ed visit or did they have to have radiologic evidence of emphysema for COPD at least? Third, generating reliable prediction models requires larger sample size. As a clinician, if I read this article as a reader, I won't be able to draw reliable conclusions or use the prediction model presented in this manuscript as it was derived from 81 patients with no validation.”

Response: 

Thank you for your detailed reflections; we have responded below to each of the points. 

1) Although SCS prescription is guideline recommended, it is clear that there is a real concern with regard to harm and a personalised approach to SCS is being evaluated (Bafadhel et al, AJRCCM 2011; Sivapalan LRM 2019). Furthermore, the evidence from Cochrane reviews heavily relies upon data from several decades and does not prevent further improvements being sought. In the real world not all patients will receive SCS at time or presentation with an exacerbation and this study is aimed at trying to determine the factors which can inform the decision to prescribe SCS and thus take a personalised individual patient approach. We have discussed these comments in the revised manuscript (page 4, lines 64-77).

2) We agree that it would appear that antibiotics prescription for asthma exacerbations are high; however, this is not an uncommon occurrence, as has previously been shown (Bafadhel et al, Chest 2011). We believe this is because patients are not seen by specialists in ED and reflects clinical practice. We have discussed this further in the manuscript and relate it to why a tool to help non-specialists is advantageous (page 18, lines 301-305). 

3) In the UK, 15% of COPD is diagnosed on admission, whilst spirometry is not performed in over 40% of patients in the community. This would lead to a large proportion of patients presenting without spirometric classification and is reflective of clinical care. Combining this in this study reflects how an algorithm could help non-specialists. This has been discussed in the manuscript (page 20, lines 343-348). 

4) Diagnosis was clinical history recorded from the patient and their medical records if available.

5) Regarding your point as to the lack of validation, in the main manuscript Discussion section we have clarified that performance of our random forest algorithms in our study was reported on test patients (see page 20, 357-359 and page 21 line 360). Performance of the models is tested on a validation set as per the supplement (page 2, lines 28-51), but we agree should be further validated in an external validation set, as per our discussion (main manuscript page 20, line 356).

Thank you again for your consideration of our manuscript.

Kind regards,

Mr Andreas Halner, Dr Richard Russell and Professor Mona Bafadhel

on behalf of the co-authors

---

## [Decision Letter · Decision Letter 2]

28 Jun 2021

Predicting Treatment Outcomes Following an Exacerbation of Airways Disease

PONE-D-20-36305R2

Dear Dr. Halner,

Both reviewers indicated "All comments have been addressed" although the first reviewer suggested "Rejection." I feel that you indeed addressed all comments and critique and therefore made the decision to accept the paper. Congratulations!

We’re pleased to inform you that your manuscript has been judged scientifically suitable for publication and will be formally accepted for publication once it meets all outstanding technical requirements.

Kind regards,

Eugene Demidenko, Ph.D.

Academic Editor

PLOS ONE

Additional Editor Comments (optional):

Reviewers' comments:

Reviewer's Responses to Questions

**Comments to the Author**

1. If the authors have adequately addressed your comments raised in a previous round of review and you feel that this manuscript is now acceptable for publication, you may indicate that here to bypass the “Comments to the Author” section, enter your conflict of interest statement in the “Confidential to Editor” section, and submit your "Accept" recommendation.

Reviewer #1: All comments have been addressed

Reviewer #2: All comments have been addressed

2. Is the manuscript technically sound, and do the data support the conclusions?

Reviewer #1: Yes

Reviewer #2: (No Response)

3. Has the statistical analysis been performed appropriately and rigorously? 

Reviewer #1: Yes

Reviewer #2: (No Response)

4. Have the authors made all data underlying the findings in their manuscript fully available?

Reviewer #1: No

Reviewer #2: (No Response)

5. Is the manuscript presented in an intelligible fashion and written in standard English?

Reviewer #1: Yes

Reviewer #2: (No Response)

6. Review Comments to the Author

Reviewer #1: 1. please capitalise 'Shapiro-Wilk' and 'Wilkcoxon; (which should replace wilcox)

2. I still think that you have not justified the analysis of the factors associated with SCS prescription. Some of these factors are likely to be confounding, and I still think that a qualitative survey of health care professionals would be better placed to answer this question. I would strongly suggest removing this component of the paper unless you are able to better justify it.

3. Similarly, I don't think you have sold me on the value of this analysis. I see that PPI as a protective factor for treatment failure is a nice finding, but you are not the first to report this - see this for example: https://jamanetwork.com/journals/jamainternalmedicine/article-abstract/227086. I need to see some discussion about the clinical implications of this research. What are you proposing should be done as a result of your study? If you are highlighting people who are at a higher risk of treatment failure (although you have not provided a decision tool to predict this either, just reported some risk factors) then what is your alternative proposal?

Reviewer #2: I salute the authors for their responses and congratulate them for this work. I feel they have provided adequate responses to my previous concerns.

7. PLOS authors have the option to publish the peer review history of their article (what does this mean?). If published, this will include your full peer review and any attached files.

Reviewer #1: No

Reviewer #2: No

---

## [Editor Report · Acceptance letter]

5 Aug 2021

PONE-D-20-36305R2 

Predicting Treatment Outcomes Following an Exacerbation of Airways Disease 

Dear Dr. Halner:

I'm pleased to inform you that your manuscript has been deemed suitable for publication in PLOS ONE. Congratulations! Your manuscript is now with our production department. 

Kind regards, 

on behalf of

Dr. Eugene Demidenko 

Academic Editor

PLOS ONE